# Microglia and amyloid precursor protein coordinate control of transient *Candida* cerebritis with memory deficits

Yifan Wu[1,2], Shuqi Du[3], Jennifer L. Johnson[4,5,6], Hui-Ying Tung[1,2], Cameron T. Landers[2,7,8], Yuwei Liu[4,5,6], Brittany G. Seman[9], Robert T. Wheeler [9,10], Mauro Costa-Mattioli [4,5,6], Farrah Kheradmand[1,2,7,11], Hui Zheng[5,6,12] & David B. Corry[1,2,7,11]

Bloodborne infections with *Candida albicans* are an increasingly recognized complication of modern medicine. Here, we present a mouse model of low-grade candidemia to determine the effect of disseminated infection on cerebral function and relevant immune determinants. We show that intravenous injection of 25,000 *C. albicans* cells causes a highly localized cerebritis marked by the accumulation of activated microglial and astroglial cells around yeast aggregates, forming fungal-induced glial granulomas. Amyloid precursor protein accumulates within the periphery of these granulomas, while cleaved amyloid beta (Aβ) peptides accumulate around the yeast cells. CNS-localized *C. albicans* further activate the transcription factor NF-κB and induce production of interleukin-1β (IL-1β), IL-6, and tumor necrosis factor (TNF), and Aβ peptides enhance both phagocytic and antifungal activity from BV-2 cells. Mice infected with *C. albicans* display mild memory impairment that resolves with fungal clearance. Our results warrant additional studies to understand the effect of chronic cerebritis on cognitive and immune function.

[1] Departments of Pathology and Immunology, Baylor College of Medicine, One Baylor Plaza, Houston, TX 77030, USA. [2] Biology of Inflammation Center, Baylor College of Medicine, One Baylor Plaza, Houston, TX 77030, USA. [3] Molecular and Cellular Biology, Baylor College of Medicine, One Baylor Plaza, Houston, TX 77030, USA. [4] Neuroscience, Baylor College of Medicine, One Baylor Plaza, Houston, TX 77030, USA. [5] Molecular and Human Genetics, Baylor College of Medicine, One Baylor Plaza, Houston, TX 77030, USA. [6] Memory and Brain Research Center, Baylor College of Medicine, One Baylor Plaza, Houston, TX 77030, USA. [7] Medicine, Baylor College of Medicine, One Baylor Plaza, Houston, TX 77030, USA. [8] Translational Biology and Molecular Medicine Program, Baylor College of Medicine, One Baylor Plaza, Houston, TX 77030, USA. [9] Molecular and Biomedical Sciences, University of Maine, Orono, ME 04469, USA. [10] Graduate School of Biomedical Sciences and Engineering, University of Maine, Orono, ME 04469, USA. [11] Michael E. DeBakey VA Center for Translational Research on Inflammatory Diseases, Houston, TX 77030, USA. [12] Huffington Center on Aging, Baylor College of Medicine, One Baylor Plaza, Houston, TX 77030, USA. Correspondence and requests for materials should be addressed to D.B.C. (email: dcorry@bcm.edu)

iverse environmental fungi are increasingly recognized as causal or contributory to the majority of common chronic, cutaneous inflammatory conditions such as atopic dermatitis (eczema), onychomycosis, and common mucosal inflammatory conditions such as pharyngitis/laryngitis, esophagitis, asthma, chronic rhinosinusitis, vaginosis, and colitis[1]. Cutaneous candidal disease in the form of mucocutanous candidiasis assumes a much more invasive and destructive character in the context of immunodeficiencies[1,2]. Fungi are further implicated in diseases as diverse as rheumatoid arthritis[3] and Alzheimer's disease (AD)[4–8].

In addition to their frequent involvement in mucosal and cutaneous diseases, the fungi are further emerging as major causes of invasive human diseases such as sepsis, especially in intensive care units in the context of critical illness. Candidemia and fully invasive candidiasis, mainly caused by *Candida albicans* and related species[9,10], is an especially serious concern in the nosocomial setting where it has emerged as one of the leading bloodstream infections in developed countries, producing high mortality and costing >1 billion dollars annually in the United States alone[11]. Diagnosis of candidemia can be difficult, as clinical signs and symptoms are often protean and non-specific, often presenting late in the course of infection when therapy is much less likely to be effective[12]. Moreover, blood fungal cultures and fungal-based serodiagnostic approaches lack sensitivity. Thus, a better understanding of fungal, especially candidal, disease pathogenesis, diagnosis, and therapy is emerging as an essential medical challenge of the 21st century.

Unique inflammatory responses have evolved to combat fungi growing along epithelial surfaces. Careful dissection of mucosal allergic inflammatory responses has revealed that characteristic granulocytes (eosinophils), cytokines (interleukin (IL)-5 and IL-13), and T effector cells (T helper type 2 (Th2) cells; Th17 cells) are potently fungicidal or at least are required for optimal fungal clearance at mucosal sites in vivo[13,14].

The rising prevalence of candidemia, often nosocomially aided through intravascular instrumentation, but also occurring as a consequence of mucosal colonization[9], raises fundamental questions regarding the physiological effect of fungal sepsis and the immune responses that are activated during disseminated disease. Fungal sepsis/hematogenous dissemination specifically does not elicit allergic responses, which instead appear to be reserved to prevent fungal dissemination from mucosal sites, and rapidly attenuate in favor of type 1 and type 17 immunity when dissemination occurs, at least in the context of hyphal fungal disease due to *Aspergillus* spp.[13–15]. In part, such fungal-immune system cross-talk involves two-way interactions with innate immune cells that specifically attenuate fungal, especially *Candida*, virulence, and regulate adaptive immunity[16,17].

Under resting conditions, the brain receives a relatively large fraction of the cardiac output (14%) and hence is susceptible to invasion due to blood-borne pathogens such as *Candida* spp. *Candida* brain infections have long been recognized as the most common cause of mycotic cerebral abscess seen at autopsy, and often present as delirium in the context of chronic illness[18,19]. Delirium is commonly seen in ICU patients who are highly susceptible to candidal sepsis, but aside from the tentative association seen between central nervous system (CNS) infection with *Candida* spp. and AD[4–8], the clinical presentation of metastatic CNS infection complicating *Candida* sepsis is poorly understood.

Experimentally, high-grade candidemia is lethal to mice and produces a profound cerebritis marked by dissemination of the organism throughout the cerebral cortex and induction of type one immunity with neutrophilia that is devoid of allergic character[20]. However, in many human contexts, candidemia resulting from a variety of pathologies is likely to be low-grade, involving

periodic showering of the CNS and other organs with relatively few organisms that may gain vascular entry from mucosal sites[16].

In this study, we sought to model the effect of low-grade, transient candidemia and *Candida* cerebritis on cerebral function and further define the major immune mechanisms involved in resolving these potentially common CNS infections. We show that hematogenously acquired *C. albicans* are readily able to penetrate the mouse blood brain barrier (BBB) and establish a transient cerebritis that causes short-term memory impairment. We further show that the cerebritis is characterized by a unique pathologic structure, the fungal-induced glial granuloma, that is marked by focal gliosis surrounding fungal cells and the deposition of both amyloid precursor protein (APP) and amyloid beta peptides, that latter which promote anti-fungal immunity. These granulomas are further accompanied by increased production of the innate cytokines IL-1β, IL-6, and tumor necrosis factor (TNF), and enhanced phagocytic capacity of microglial cells. Thus, even low-grade candidemia can produce a physiologically significant brain infection.

## Results

**Acute model of intracranial fungal infection.** To begin to understand the CNS effects of low-grade, transient fungemia, we developed a mouse model of fungal cerebritis based on the model developed by Lionakis et al.[20]. We developed this model using *C. albicans* as this organism is one of the most common fungi isolated from human blood[9] and is a significant cause of human CNS infection[21]. Intravenous injection of large numbers of *C. albicans* (e.g., $10^5$–$10^6$ organisms) induces considerable mortality in mice[20]. To avoid this and mimic more accurately the transient and silent fungemias that are likely to occur in humans, we modified this model to include fewer organisms (2500–50,000 yeast cells). We discovered that a single injection of 25,000 viable cells of *C. albicans* into wild-type C57BL/6 mice produced a transient cerebral infection that was detectable 4 days post challenge, but largely cleared by day 10 (Fig. 1a, b). This degree of infection produced no fever or hypothermia (Supplementary Figure 1A), no obvious abnormal behavior, and no mortality. We found no evidence pathologically or behaviorally that *C. albicans* induced meningitis as a result of our intravenous challenge protocol.

To determine where in the brain the fungi dispersed after hematogenous administration, we performed immunofluorescence staining on coronal whole brain sections from infected mice 4 days post i.v. injection. We discovered multiple (~5–10/brain), discrete, roughly spherical lesions, ~50–200 μm in diameter occurring in both cerebral hemispheres, but sparing the cerebellum, that consisted of the central accumulation of cells that avidly stained for periodic acid-Schiff (PAS), a general marker of polysaccharides, containing small nuclei as assessed by DAPI (Fig. 1c–e). These lesions further consisted of the focal aggregation of astrocytes, assessed as GFAP-expressing cells, and microglia, assessed as IBA1-expressing cells, surrounding the PAS-positive central cells (Fig. 1c–h). The focal astrocytosis and microgliosis consisted of a rim of aggregated cells that did not enter the central areas (Fig. 1f–h). Additional staining by calcofluor white, which binds specifically to the fungal cell wall polysaccharide chitin[22], confirmed that at the center of these lesions were numerous yeast cells (Fig. 1i).

In contrast to the more diffuse, disseminated CNS lesions that result with high-grade *C. albicans* challenges[20], these subclinical fungal infections induced significant recruitment of cerebral monocytes, but not neutrophils as demonstrated by flow cytometry[20] (Supplementary Fig. 1B). We further did not observe the conversion of *C. albicans* yeasts into hyphal forms, a marker

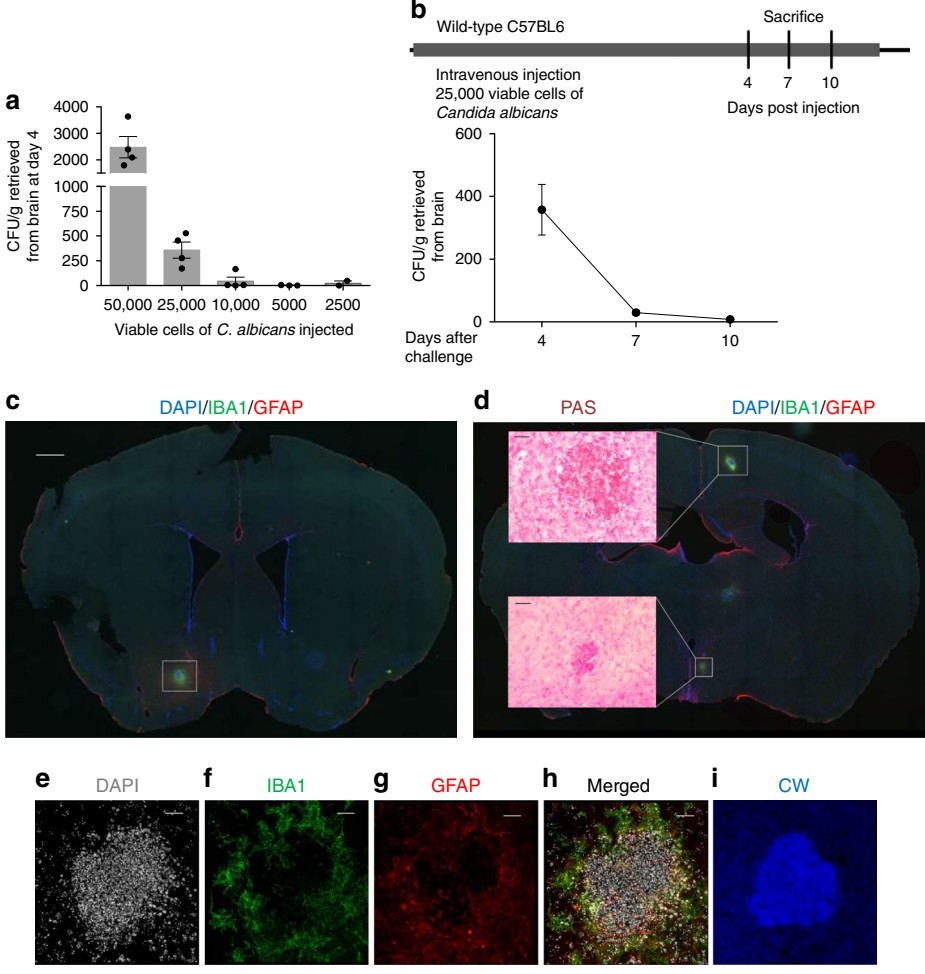

**Fig. 1** Brain recovery of *Candida albicans* after intravenous injection. **a** Wild-type C57BL/6 mice were challenged once intravenously with the indicated doses of *C. albicans* and the mean colony forming units (CFUs) cultured from whole brain 4 days later were determined. **b** Clearance of *C. albicans* from brains of mice after i.v. challenge with 25,000 viable cells over 10 days. **c**, **d** Immunofluorescent staining of mouse brain 4 days post i.v. challenge. **c** Low-magnification view of whole brain after staining with DAPI, IBA1, and GFAP (scale bar: 750 μm) and **d** PAS staining on a consequent slide demonstrated granuloma-like structures (scale bar: 20 μm). **e–i** Higher magnification views of the boxed area in **c** stained with DAPI, IBA1, GFAP, and calcofluor white (CW), respectively. The merged image is shown in **h**. (Scale bar: 20 μm. $n = 4$ in **b**). Data are representative of three (**a**, **b**) and six (**c–i**) independent experiments

of more aggressive infection[23], as was seen in fatal invasive *Candida* cerebritis, nor did we observe fungal forms or lymphocytes outside the areas of gliosis (Fig. 1d and data not shown)[20]. Together, these observations describe a new type of CNS lesion arising in the context of subclinical fungal infections in which the organism is tightly contained in areas of gliosis. We term this novel type of focal inflammatory process due to CNS fungi a fungal-induced glial granuloma (FIGG).

**FIGGs are linked to activated microglia and innate cytokines**. We conducted additional studies to understand the cellular and biochemical inflammatory accompaniments of FIGGs. We first compared IBA-1 stained coronal sections of brains from mice injected with fungi or sham. Within FIGGs, we observed hypertrophic microglia that stained brightly for IBA-1 (IBA-1_high), indicative of microglial activation and proliferation[24,25], as compared to brain from sham-infected mice (Supplementary Fig. 2). IBA-1_high cells were not found in any brain sections of control mice (data not shown). Enumeration of all and IBA-1_high microglia from coronal sections of different brain regions revealed increased numbers of both total and especially activated

microglia, consistent with prior findings of microgliosis in association with brain inflammation (Supplementary Fig. 2)[24].

The transcription factor nuclear factor kappa B (NF-κB), which comprises a small family of functionally distinct transcription factors, is commonly activated in immune contexts, including during fungal infections where it is required to activate effective anti-fungal immunity[26]. We assessed induction of both NF-κB messenger RNA (mRNA) and protein (p65 subunit) in total mRNA and protein extracted from brain between 4 and 14 days following i.v. challenge with 25,000 *C. albicans* cells (Fig. 2a–c). Relative to naive brains, we found that NF-κB p65 expression was significantly elevated at both RNA and protein levels at all timepoints examined, with the highest levels seen at day 14, several days past the point at which infection was no longer detectable (Fig. 1b). These findings were confirmed by demonstrating the downregulation of the inhibitory NF-κB subunit IκBα under the same conditions (Supplementary Fig. 3).

Among many genes induced by NF-κB are the pro-inflammatory innate immune cytokines IL-1β, IL-6, and TNF[27]. These cytokines were profoundly induced in the brains of mice at most or all times examined after fungal challenge through day 14, well after the point at which fungi could no longer be cultured

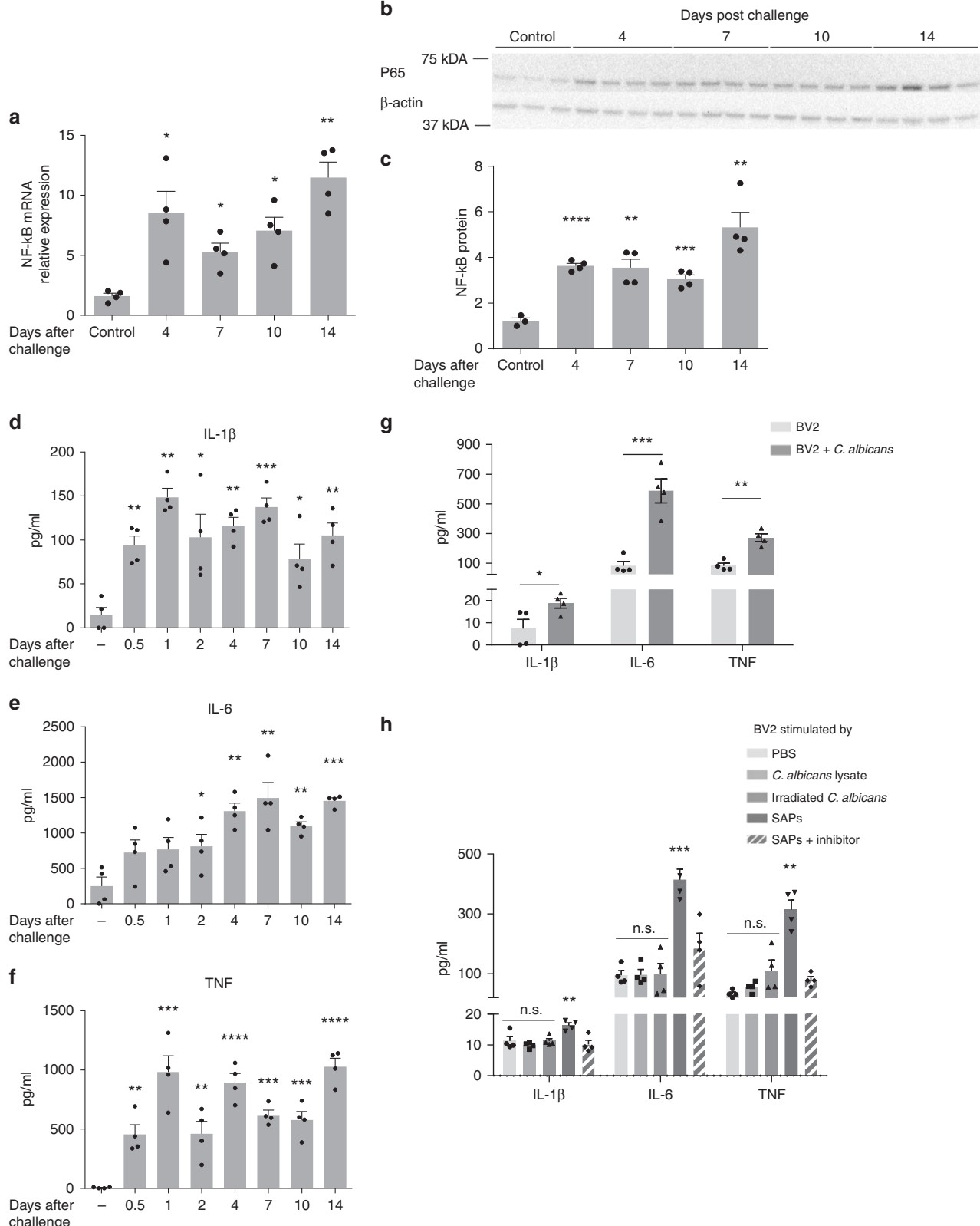

from brains (Figs. 2d–f; 1b). Microglia are known to produce these cytokines in the context of other CNS inflammatory diseases such as AD[28,29]. We hypothesized that microglia inducibly secrete IL-1β, IL-6, and TNF in response to *C. albicans*. To test this, we utilized the immortalized murine microglial cell line BV-2. Co-culturing BV-2 cells ($1 \times 10^5$/ml) with 200 viable cells of *C. albicans* led to significantly enhanced secretion of IL-1β, IL-6, and TNF as compared to control-treated cells (Fig. 2g). Of note, neither whole *C. albicans* lysate antigen equal to 200 viable cells nor irradiated *C. albicans* induced any cytokine

**Fig. 2** *C. albicans* induces pro-inflammatory cytokines in whole-brain and BV-2 cells. C57BL/c mice were challenged i.v. with 25,000 CFU of *C. albicans* after which whole brains were harvested at the indicated days for analysis by real-time quantitative PCR (RT-qPCR), western blot, or ELISA. **a, b** Nuclear factor kappa B (NF-κB) expression as assayed by RT-qPCR (**a**) or western blot (**b**; p65 subunit) over 14 days. **c** Densitometric analysis of the data from **b**. **d–f** IL-1β, IL-6, and TNF cytokine levels from whole-brain homogenates as assessed by ELISA. **g** BV-2 cells were seeded for 6 h in 24-well plates ($1 \times 10^5$ cells per well) and then incubated with *C. albicans* (200 viable cells per well) for 16 h after which secreted IL-1β, IL-6, and TNF were quantitated by ELISA. **h** BV-2 cells were seeded same as above and incubated with lysates of *C. albicans* (equivalent 200 viable cells per well), irradiated *C. albicans* (200 cells per well), secreted aspartic proteases (SAP, 1 μM), or inhibited secreted aspartic proteases (1 μM) for 16 h and the same cytokines were quantitated by ELISA ($n = 4$. mean ± S.E.M, ns: not significant, $*p < 0.05$, $**p < 0.01$, $***p < 0.001$, $****p < 0.0001$, using two-tailed Student's *t*-test (**g**) or one-way ANOVA (**a**, **c–f**, **h**) followed by Dunnett's test for multiple comparison). Data are shown as representative of two independent experiments

secretion, whereas purified secreted aspartic proteinases (SAPs) stimulated the production of these pro-inflammatory cytokines as previously described in a manner that required intact proteinase activity[30,31] (Fig. 2h). Thus, microglial cells upregulate the production of the type 1 cytokines IL-1β, IL-6, and TNF in the presence of viable fungal cells, possibly via SAPs.

**APP is upregulated in FIGGs during *C. albicans* cerebritis**. One of the hallmarks of chronic brain inflammation and degeneration (e.g., AD) is the presence of parenchymal plaques composed of amyloid β (Aβ) peptides that are cleaved from APP[32]. Studies have suggested that β-amyloid peptides possess anti-microbial activity[33,34], especially against *C. albicans*. We sought to determine if in the context of *C. albicans* cerebritis brain cells increase production of APP. To address this, we first performed quantitative PCR on total RNA extracted from brains of mice at different days post infection with *C. albicans* (Fig. 3a). Relative to control brains, we found significantly higher expression of APP mRNA at all timepoints examined after infection out to 10 days. We also measured APP by western blot from the same brains and confirmed a progressive, more than threefold increase in APP production by day 14 following infection with *C. albicans* (Fig. 3b, c).

To determine where in the brain enhanced production of APP was occurring, we performed fluorescent immunohistochemistry for APP on sections of brain from wild-type mice 4 days after infection with *C. albicans* using an antibody that only recognizes APP and not Aβ. Although APP is widely expressed in brains, we found that enhanced accumulation of APP following *C. albicans* challenge occurred almost exclusively around FIGGs, and primarily within the areas of gliosis (Fig. 3d), observations that were confirmed through additional side-by-side staining of comparable FIGGs from *app*[−/−] mice, in which no APP signal was detected (Fig. 3e). These results demonstrate that APP is upregulated in the brain and accumulates around FIGGs in areas of gliosis, but sparing the central areas that contain the fungi, during acute *C. albicans* cerebritis. Of note, the general appearance of FIGGs did not differ between wild-type and *app*[−/−] mice.

**Aβ physically associates with *C. albicans* within FIGGs**. We next addressed whether cleaved amyloid β peptides localize similarly to APP using a peptide-specific antibody. Surprisingly, we found that in contrast to the distribution of APP, amyloid β peptides were concentrated in the center of the FIGGs, presumably in direct contact with the fungal cells (Fig. 4a). To again validate the specificity of the antibody against amyloid β peptides, we also carried out side-by-side control staining for amyloid β peptides in brain sections from infected *app*[−/−] mice and again no signal was observed (Fig. 4b). We further determined by ELISA that soluble amyloid β peptides are persistently elevated in mouse brains well past fungal challenge and clearance (Fig. 4c, d).

Of note, insoluble amyloid β aggregation was not observed via thioflavin S staining in this acute infectious model (data not shown).

APP is cleaved endogenously to yield amyloid β peptides by the peptidases β secretase (BACE-1) and presenilin 1 (PS1)[35,36]. To further support the molecular link between amyloid β peptides and *C. albicans* cerebritis, we measured β and γ secretase levels from mouse brain homogenates and observed a significant increase in the protein levels of BACE-1 and PS1, which is a subunit of γ secretase (Supplementary Fig. 4). Thus, in contrast to APP, which localized only to areas of gliosis, amyloid β peptides localized to fungal cells exclusively within the center of FIGGs through a *C. albicans*-dependent mechanism that involves the induction of β and γ secretases.

**Aβ promotes anti-fungal immunity by stimulating BV-2 cells**. Previous studies have shown that Aβ peptides interact physically with *C. albicans* in vitro and may be directly fungistatic[33,34,37]. We first attempted to verify that Aβ peptides possess anti-fungal activity in vivo. 5xFAD mice that overexpress human APP in the cerebrum and *app*[−/−] mice were challenged i.v. with 25,000 viable cells of *C. albicans* and brains were removed at different days post challenge for fungal recovery. We found that clearance of *C. albicans* at days 4 and 7 from mouse brain was strongly impaired by the lack of APP, but conversely was markedly enhanced in 5xFAD mice (Fig. 5a). Nonetheless, all mice in this experiment achieved brain sterility by day 10 (Fig. 5a). *app*[−/−] mice further developed significant hypothermia, a sign of overwhelming infection, while demonstrating significantly impaired secretion of pro-inflammatory brain cytokines at days 4 and 7 (Supplementary Fig. 5B). These results indicate that APP or Aβ peptides promote anti-fungal immunity at early timepoints after infection.

We next utilized an in vitro fungistasis assay[38] to determine if Aβ peptides possess anti-fungal (either fungicidal or fungistatic) activity (Fig. 5b). This assay involves the microscopic enumeration of growing *C. albicans* colonies in response to Aβ peptides, with or without addition of BV-2 cells in comparison to control conditions. In keeping with prior studies[34], we initially incubated *C. albicans* with 50 μg/ml of mouse amyloid β peptides or scrambled control peptide to determine if Aβ peptides were directly fungistatic. In contrast to prior observations[33,34], we found no inhibition of *C. albicans* growth under these assay conditions (Fig. 5c).

We next modified the fungistasis assay by adding BV-2 cells that had been pre-stimulated with and Aβ peptides or control to determine if Aβ peptides can indirectly induce fungistasis through bystander cells. We found that at concentrations of 1 μM (4 μg/ml), both Aβ 1–40 and 1–42, but not scrambled peptide, aggregated tightly around yeast cells as previously described[34,37]. More importantly, both Aβ1–40 and 1–42, but not scrambled peptide, significantly stimulated fungistasis when pre-incubated

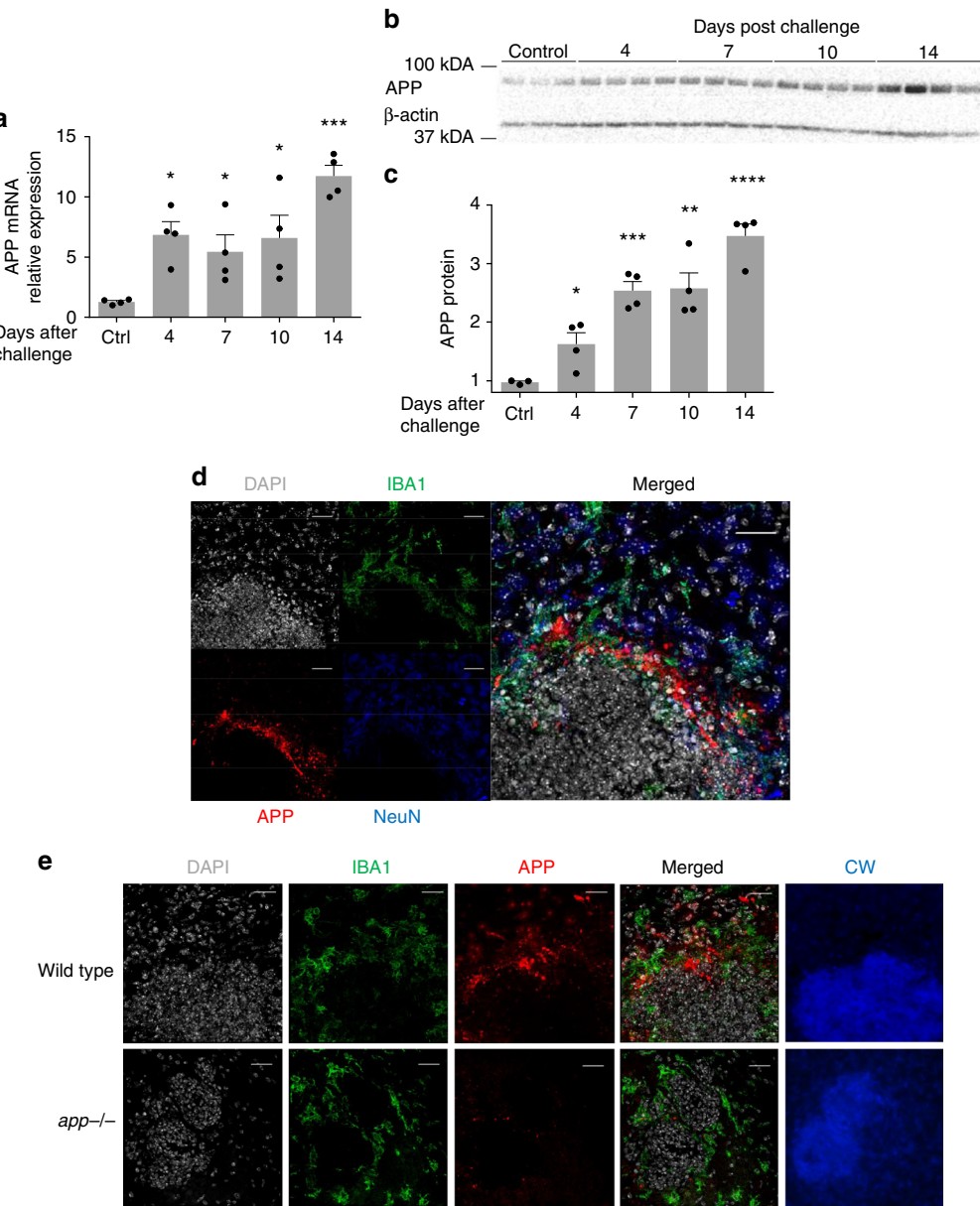

**Fig. 3** *C. albicans* cerebritis induces upregulation of amyloid precursor protein (APP). Mice were challenged with *C. albicans* as in Fig. 2 and RNA and protein extracted from whole brain. **a** qRT-PCR was performed to quantify mRNA for APP over 14 days. **b** Western blot analysis of APP over the same time period. **c** Densitometric analysis of the western blot data. **d** Immunofluorescence images of brain from *C. albicans*-infected wild-type mice 4 days post i.v. challenge with *C. albicans* showing staining for DAPI, IBA1, APP, NeuN, and merged images. Images are centered on fungal granulomas similar to those shown in Fig. 2. **e** Repeat staining of fungal granulomas for DAPI, IBA1, APP, and calcofluor white (CW) comparing wild-type to *app*−/− mice. (*n* = 4, mean ± S.E.M, *\*p* < 0.05, *\*\*p* < 0.01, and *\*\*\*p* < 0.001 using one-way ANOVA followed by Dunnett's test for multiple comparison. Scale bar: 20 μm). Data are shown as representative of two independent experiments

with BV-2 cells prior to the addition of yeast cells to the assay (Fig. 5d). Of note, human Aβ peptides induced similar fungistatic activity in BV-2 cells (Supplementary Fig. 6A).

To further define the mechanisms by which APP-derived peptides induce fungistasis, we first performed a supernatant transfer experiment in which BV-2 supernatants were collected after pre-stimulation with Aβ peptides and transferred to monocultures of *C. albicans* (Fig. 5e). We discovered that the Aβ peptide-stimulated, BV-2 cell-free supernatant was sufficient to induce significant fungistasis as compared to scrambled peptide in a manner that differed marginally from cultures having BV-2 cells present (Fig. 5d, e). These data demonstrate that whereas Aβ peptides fail to exhibit direct anti-fungal activity,

they do stimulate BV-2 cells to secrete one or more soluble anti-fungal factor.

We confirmed this stimulatory effect of Aβ peptides on microglia by pre-incubating BV-2 cells with Aβ peptides and then washing them to remove unbound peptides before addition of *C. albicans*. We observed a similar fungistatic effect from these primed BV-2 cells, further supporting the ability of Aβ peptides to activate microglia to an enhanced anti-fungal state.

**Aβ peptides enhance phagocytic activity of BV-2 cells.** Another potential mechanism by which BV-2 cells might inhibit fungi is through phagocytosis and intracellular killing[39]. To explore this

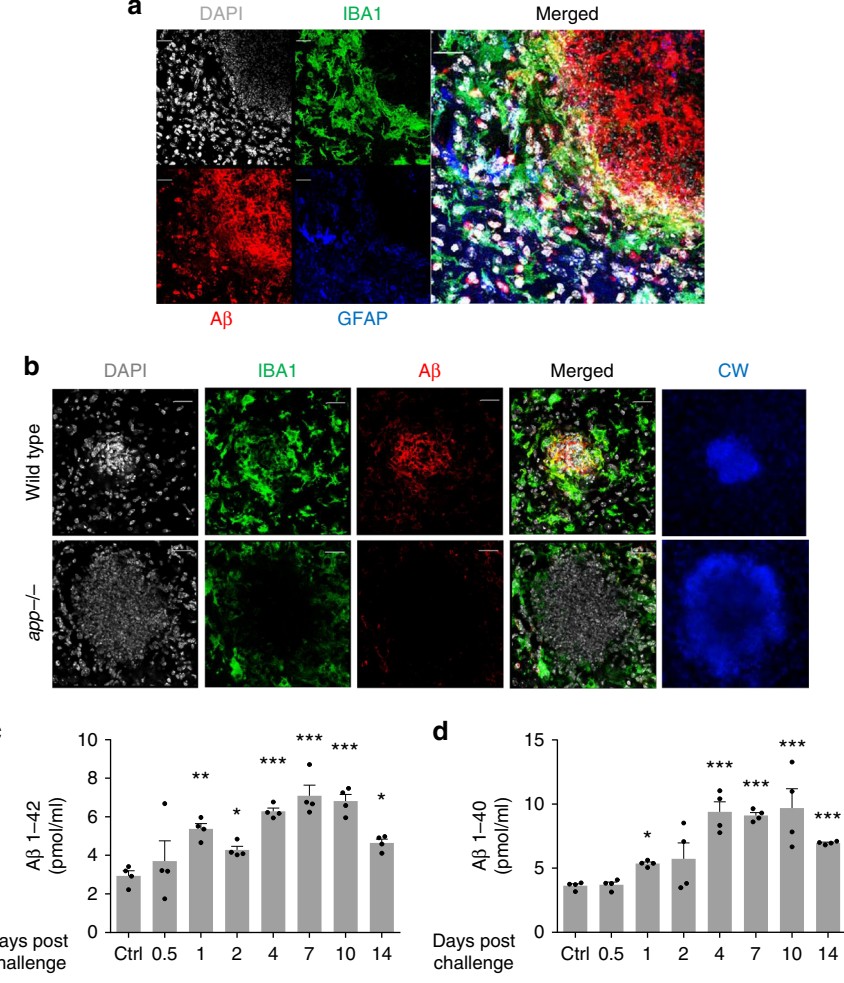

**Fig. 4** Amyloid β binds directly to *Candida albicans* in vivo. **a** Wild-type mice were challenged with *C. albicans* as in Fig. 2, and immunofluorescence staining for DAPI, IBA1, Aβ, and GFAP was performed on brain sections containing fungal granulomas. **b** Repeat staining of fungal granulomas for DAPI, IBA1, Aβ, and calcfluor white (CW) comparing wild-type to $app^{-/-}$ mice. **c, d** Aβ 1–42 and 1–40 levels from whole-brain homogenates as assessed by ELISA. ($n = 4$, mean ± S.E.M, $*p < 0.05$, and $**p < 0.01$ using one-way ANOVA followed by Dunnett's test for multiple comparison. Scale bar: 20 μm). Data are shown as representative of three independent experiments

possibility, we co-cultured BV-2 cells pre-stimulated with Aβ or scrambled peptide with fluorescent (mNeonGreen-expressing) *C. albicans* cells and determined the number of mNeonGreen$^+$/CD11b$^+$cells as a means of determining cell-yeast interactions. We found that cell-yeast interactions occurred in 21.9 ± 3.0% of control peptide-stimulated BV-2 cells, but that Aβ peptides stimulated significantly more such interactions (37.2 ± 2.8% and 41.0 ± 3.2%, Aβ40 versus Aβ42, respectively, $p < 0.05$, Fig. 6a, b). To further characterize these events, we customized a gating strategy to orthogonally compare yeast-BV-2 cell interactions as a means of determining true yeast uptake by phagocytosis (producing overlapping images) from mere cell surface association (producing non-overlapping images) (Fig. 6c). This analysis confirmed that both Aβ peptides significantly enhanced phagocytosis of *C. albicans* as compared to control-treated cells (scrambled peptide: 14.0 ± 2.0% overlap; Aβ 40 peptide: 25.0 ± 2.1% overlap; Aβ 42 peptide: 25.7 ± 2.4% overlap, $p < 0.01$; Fig. 6d).

Finally, we confirmed that phagocytically active (CD68$^{+40}$) microglial cells cluster densely at the center of FIGGs, in the immediate vicinity of the yeast clusters (Fig. 6e). Together, these findings indicate that in addition to enhancing secretion of

soluble anti-fungal factors, Aβ peptides also promote the direct phagocytic uptake and intracellular killing of yeast cells by microglia.

**Dectin-1 promotes microglial phagocytosis of *C. albicans*.** Disease-associated microglia (DAM) are recently described, highly activated microglial cells that surround inflammatory lesions in AD and other neurodegenerative disorders[41,42]. Among many inflammatory markers, DAMs express Dectin-1/Clec7A, a pattern recognition receptor expressed by phagocytic cells that recognizes fungal β-glucan[43]. We confirmed that Dectin-1 is highly expressed on DAMs of FIGGs (Fig. 7a). To demonstrate if Dectin-1 is required for the Aβ peptide-enhanced phagocytosis by microglia, we assessed by flow cytometry the uptake of fluorescent *C. albicans* with and without addition of a blocking anti-Dectin-1 antibody. We found that Dectin-1 blockade inhibited by up to 50% the phagocytoic uptake of mNeonGreen$^+$ yeast cells both at rest and after stimulation by Aβ 1–42 peptide (Fig. 7b–d). Thus, Dectin-1 expression is enhanced on DAMs associated with FIGGs and enhances the phagocytic uptake of *C. albicans* by microglial cells.

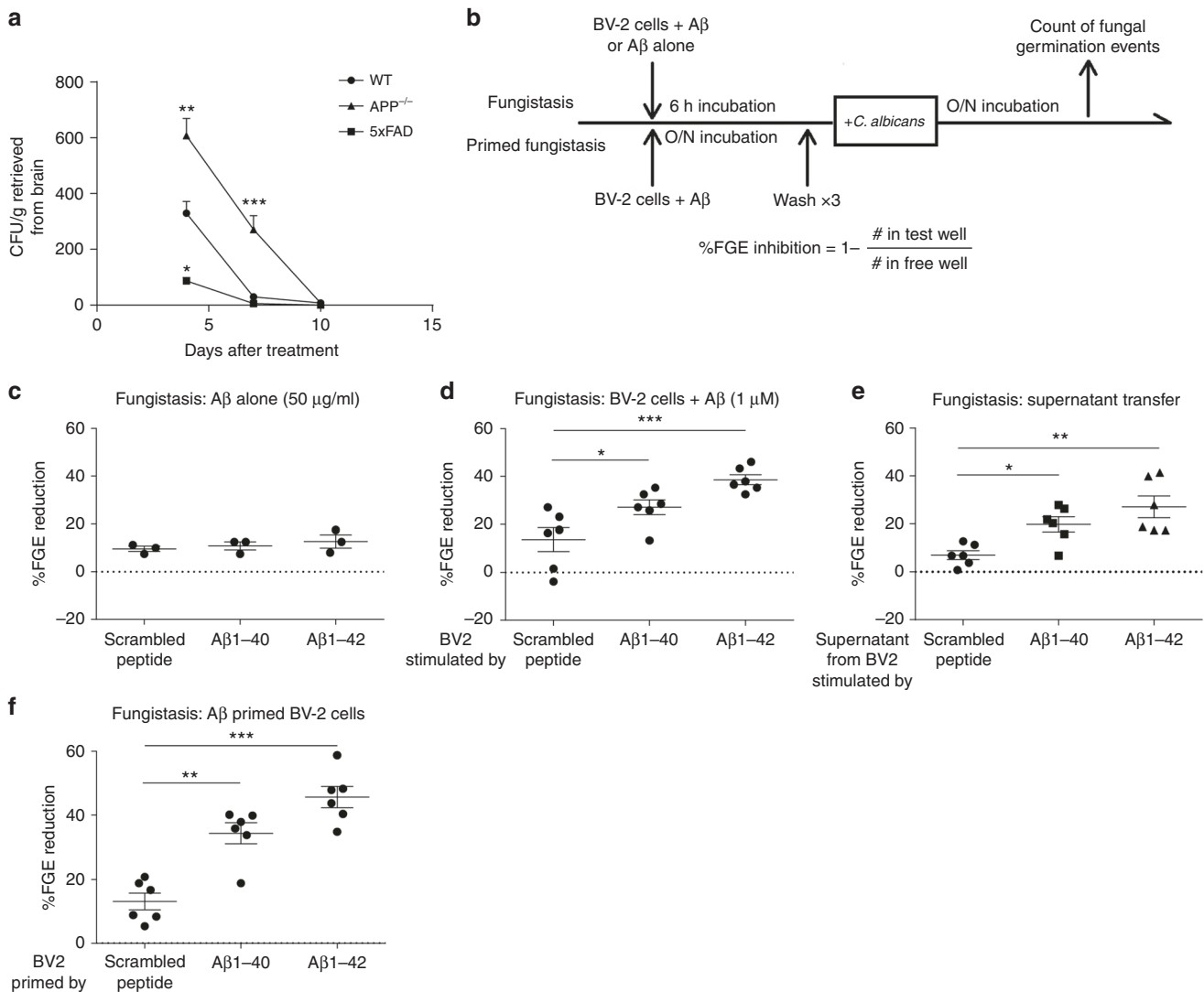

**Fig. 5** Amyloid β indirectly induces fungistasis of *Candida albicans*. **a** Clearance of *C. albicans* from brains of wild-type, *app*⁻/⁻, or 5xFAD mice after i.v. challenge with 25,000 viable cells over 10 days. **b** Schematic of in vitro fungistasis assay. **c** Aβ peptides 40 and 42 and scrambled peptide (50 μg/ml) were added to in vitro cultures of *C. albicans* (200 viable cells/ml) for 16 h and the effect on fungal growth inhibition as determined was assessed by percent fungal germination event (%FGE) inhibition. **d** BV-2 cells were pre-treated with the above peptides at 2 μg/ml for 6 h and then *C. albicans* (200 viable cells/ml) were added. Fungal inhibition was calculated same as above. **e** Supernatant from the medium in which BV-2 cells were pre-treated with Aβ peptides (**d**) was extracted and then *C. albicans* (200 viable cells/ml) were added and incubated for 16 h. **f** BV-2 cells were primed with Aβ peptides overnight and then washed three times to remove soluble Aβ peptides before addition of yeast cells. (*n* = 3 for **c**, *n* = 4 for **a**, *n* = 6 for **d**–**f**, mean ± S.E.M, \**p* < 0.01, \*\**p* < 0.01, \*\*\**p* < 0.001 using two-tailed Student's *t*-test (**a**) or one-way ANOVA followed by Tukey's test for multiple comparison (**b**–**e**). Data are shown as representative of four independent experiments.)

**Low-grade fungal cerebritis transiently impairs memory**. We carried out well-established tests of rodent behavior using naive and *C. albicans*-infected wild-type mice to determine if substandard performance in any of these assays could be correlated with fungal cerebritis. We first conducted open-field tests to quantify the degree of locomotor activity and anxiety potentially related to fungal infection that could spuriously influence subsequent behavior tests. (Fig. 8a–e). No significant differences were found in any of the five indices, suggested that mice were not experiencing severe stress following infection with *C. albicans* and are consistent with our empiric observation that mice were grossly normal following fungal infection.

We next performed T-maze spontaneous alternation tests in sham and *C. albicans*-infected mice. Compared to sham, *C. albicans*-infected mice made significantly fewer alternations, and

recovered as the infection cleared by day 10 (Fig. 8f). This result demonstrated that intracranial infection induces impaired working spatial memory. However, no difference in contextual fear conditioning, a form of associative learning and memory, was observed (Fig. 8g). Thus, acute, low-grade *C. albicans* cerebritis induces a transient, mild working memory deficit that is reversible with clearance of the infection.

## Discussion

Although recognized as an increasingly common medical problem, the long-term health effects of transient candidemia are almost completely unknown. High-grade candidemia is rapidly lethal to both humans and mice[10,20,44]. Here, however, we sought to understand how low-grade candidal sepsis affects a critical

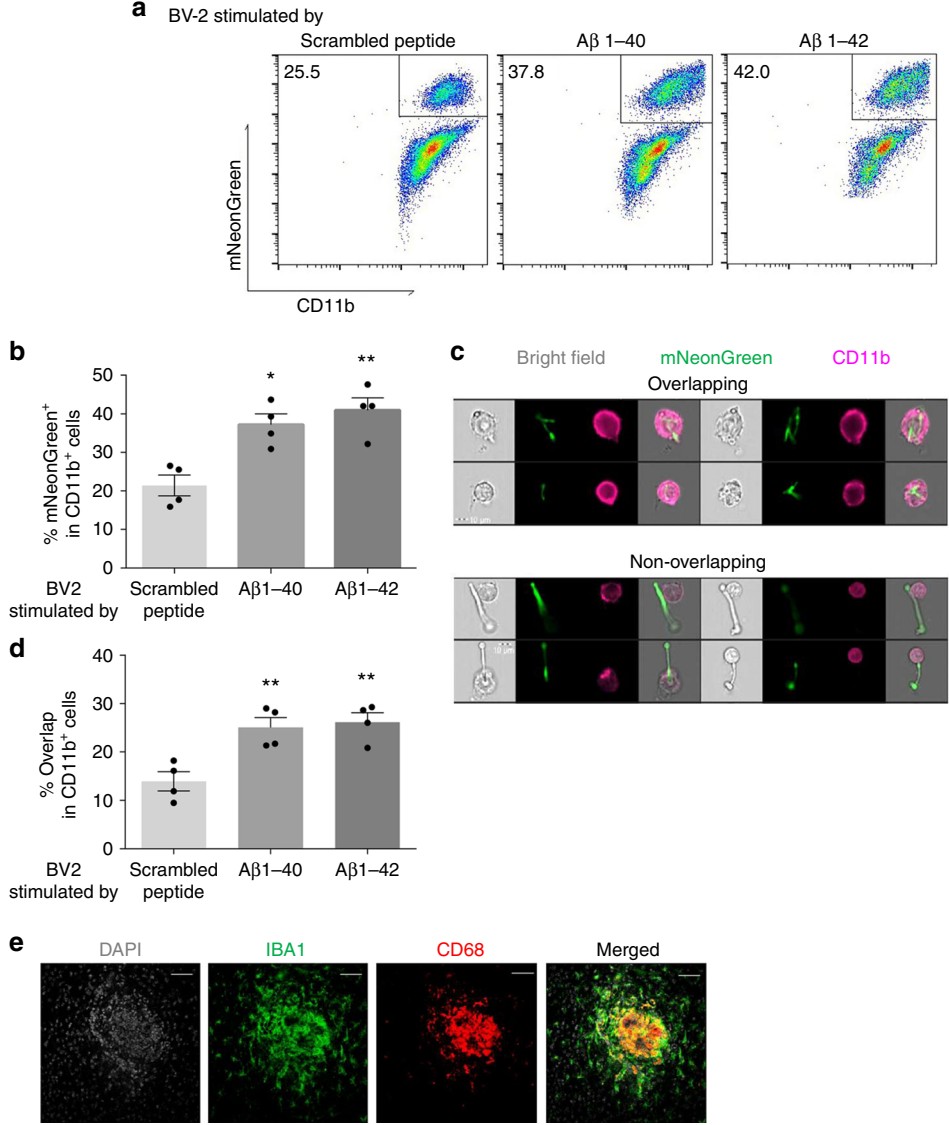

**Fig. 6** Aβ peptides enhance phagocytosis by BV-2 cells. **a** Representative flow cytometry plot (ImageStream X) of Aβ−stimulated BV-2 cells co-cultured with mNeonGreen⁺ *Candida albicans*. **b** Percentage of mNeonGreen⁺ cells in all CD11b⁺ cells. **c** Representative flow cytometry plots depicting overlapping and non-overlapping associations of yeast and CD11b⁺ cells that distinguish cell–cell adherence from phagocytosis. **d** Percentage of overlap cells in all CD11b⁺ cells. **e** Fluorescent microscopic images of a typical FIGG expressing CD68⁺/IBA1⁺ microglia ($n = 4$, mean ± S.E.M, *$p < 0.05$, **$p < 0.01$ using one-way ANOVA followed by Dunnett's test for multiple comparison. Data are shown as representative of two independent experiments)

target organ, the brain. We discovered that the intravenous injection of 25,000 viable *C. albicans* cells is surprisingly well tolerated in young, healthy mice, producing no gross abnormalities either acutely or chronically. However, a brain parenchymal infection is clearly established, albeit transiently, involving the cerebral cortices exclusive of the cerebella and meninges. Such infection further induces a robust innate immune response characterized by focal gliosis and monocytosis devoid of neutrophils or lymphocytes and marked by the production of the innate cytokines IL-1β, TNF, and IL-6, and the enhanced expression of multiple microglial activation markers that enhance phagocytic function. Such innate inflammation is ultimately successful in resolving the cerebritis, but we detected a transient decline in cerebral function that was likely due to the direct effects of the fungi and the sterilizing inflammation directed against them. These findings expand our knowledge of the CNS effects of transient candidemia and have important implications for

understanding the potentially broader role of fungi in CNS disease.

Our findings demonstrate that the CNS and systemic immune and physiologic responses to low-grade candidemia (25,000 yeast cells) differ substantially from high-grade exposures (>250,000 yeast cells). Whereas high-grade candidemia was rapidly and completely fatal and accompanied by massive cytokine release in the context of CNS neutrophilia and monocytosis, low-grade disease yielded no mortality or neutrophilia. Perhaps most strikingly, rather than diffuse CNS spread of the organism through the brain cortices as was seen in high-grade disease, low-grade candidemia yielded a strikingly focal CNS infection in which numerous organisms were collected in a unique neuro-pathologic structure that we term FIGGs. The size and spherical structure of FIGGs containing geographically distinct layers of inflammatory cells with the pathogen centrally located is highly reminiscent of more typical granulomas containing histiocytes

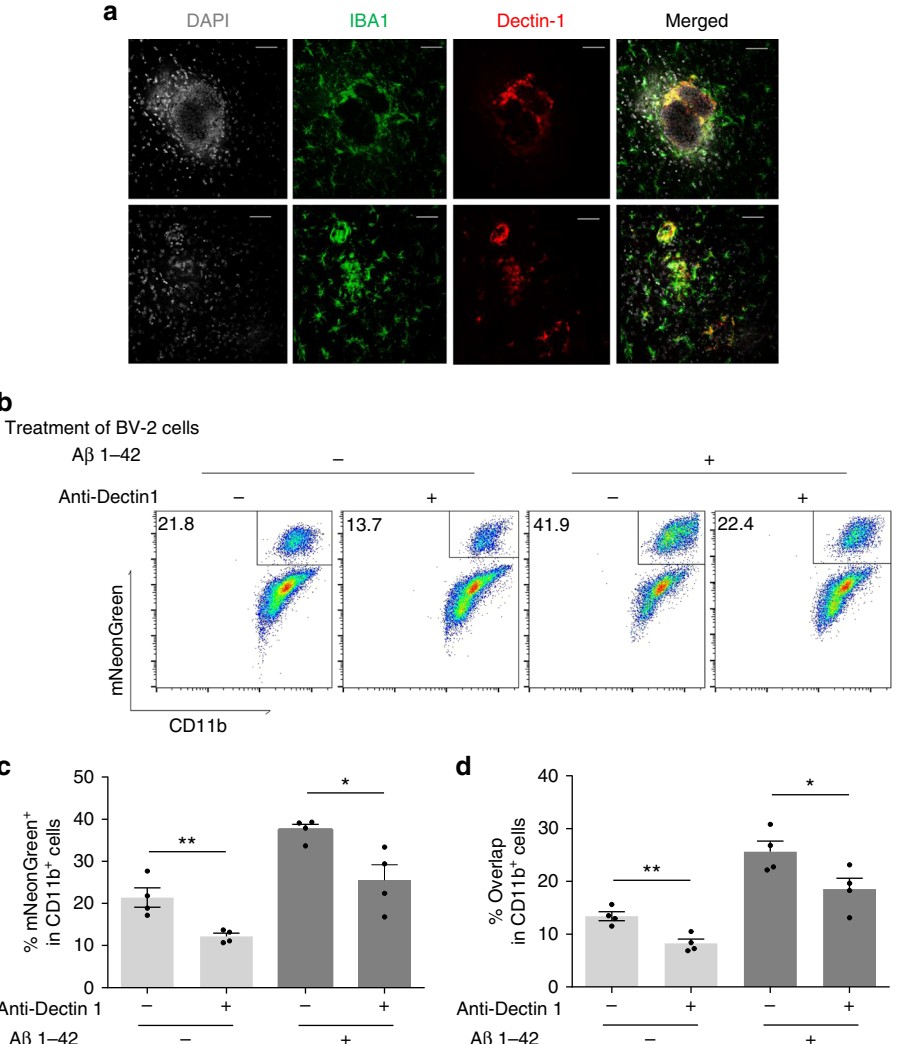

**Fig. 7** Dectin-1 expression is associated with FIGGs. **a** Wild-type mice were challenged with *C. albicans* as in Fig. 2, and immunofluorescence staining for DAPI, IBA1, and Dectin-1 was performed on brain sections containing FIGGs. **b** Representative flow plot from the ImageStream X of treated BV-2 cells (Aβ or Dectin-1 blocking antibody) co-cultured with mNeonGreen+ *C. albicans*. **c** Percentage of mNeonGreen+ cells in all CD11b+ cells. **d** Percentage of overlap cells in all CD11b+ cells with the same gating strategy in Fig. 6. (*n* = 4, mean ± S.E.M, *$p$ < 0.01, **$p$ < 0.01 using one-way ANOVA followed by Dunnett's test for multiple comparison. Data are shown as representative of two independent experiments.)

and lymphocytes that have long been known to form around fungi and other organisms in non-neuronal tissues[45].

The CNS is uniquely protected from toxic and microbial challenges by the BBB, hence it is surprising that hematogenously acquired *C. albicans* could readily pass the BBB to proliferate in the brain parenchyma of our mice[46]. The BBB is capable of halting the spread of some bacteria and many viruses to the CNS, but the unique ability of yeast cells to penetrate endothelia in vivo and in vitro[47], perhaps through elaboration of unique virulence factors such as proteinases and lytic peptides such as candidalysin[31,48], suggests that the main defense of the CNS against blood-borne fungal, especially candidal, invasion is immunologic, not physical. Additional studies are required to understand the microbial receptors and fungal virulence factors that coordinate such effective protection.

In addition to efficiently sterilizing the brain following low-grade *Candida* infection, the innate immune response elicited appears to also attenuate the pathogenicity of the organism. Unrestrained *Candida* yeast forms acquire tissue-invasive potential as marked by extension of pseudohyphae and full hyphal transformation as was seen in overwhelming CNS disease[16,20]. As we observed only yeast forms with no evidence of hyphae in the brains of our mice, we presume that the immune response to the *Candida*, likely including phagocytic immune cells such as microglia, precluded or reversed such pathologic transformation[49].

As histiocytic granulomas are required for optimal eradication of pathogens in the periphery[45], we presume that FIGGs are similarly essential to the rapid clearance of *C. albicans* from the CNS in our model. This is supported by our demonstration for the first time that microglia are inducibly activated into a fungicidal/fungistatic state as demonstrated by two processes, secretion of anti-fungal substances and enhanced phagocytosis. Another unique aspect of FIGGs is the geographic separation of the parent protein APP, confined to the FIGG periphery, and Aβ peptides, located centrally and physically associating with the fungi. Our data support the possibility that APP is cleaved through the activity of β and γ secretases, but also raise the intriguing possibility that APP could also be cleaved by secreted fungal proteinases. Although we could not confirm that Aβ

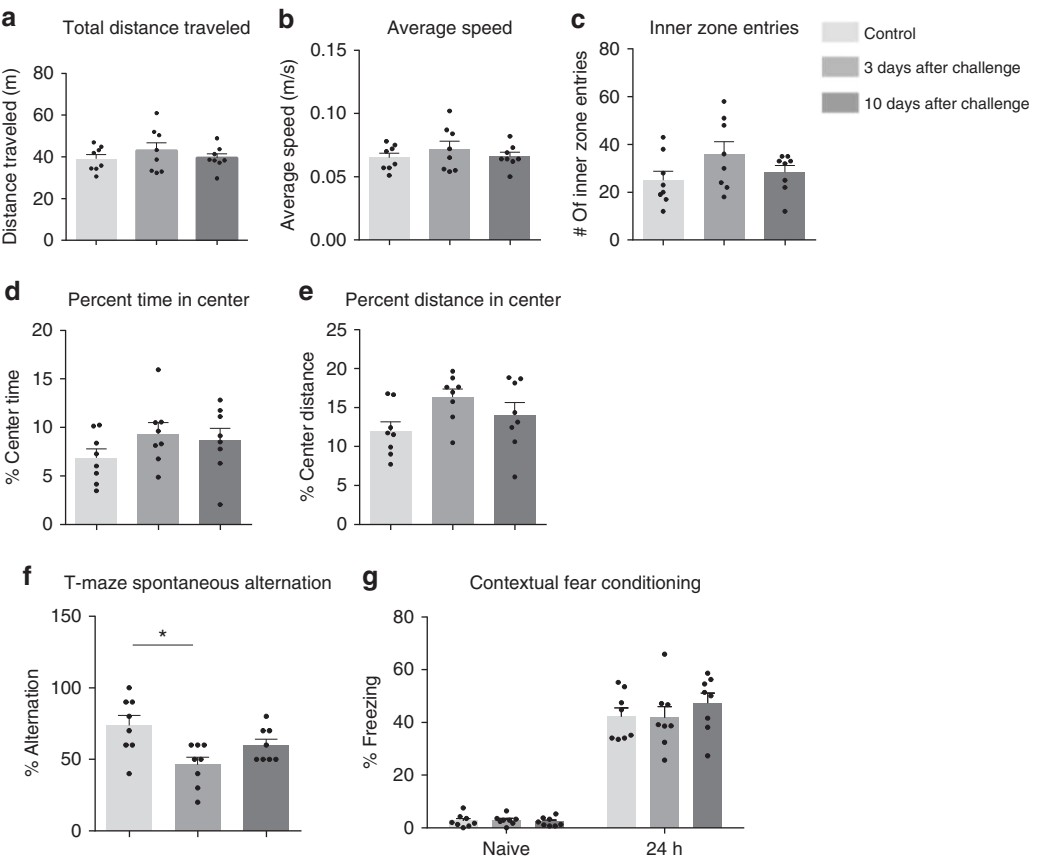

**Fig. 8** The effect of *C. albicans* cerebritis on mouse behavior. **a**–**e** After the intravenous injection of 25,000 viable cells of *C. albicans* into wild-type C57BL/6 mice, open-field tests were conducted on days 3 and 10 post fungal challenge to determine total distance traveled (**a**), average speed (**b**), inner zone entries (**c**), percent time in center (**d**), and percent distance in center (**e**). Mice were subsequently assessed in the T-maze spontaneous alternation task (**f**) and contextual fear conditioning (**g**). ($n = 8$, mean ± S.E.M, *$p < 0.05$, using one-way ANOVA followed by Dunnett's test for multiple comparison). Data are shown as representative of two independent experiments

peptides are directly candidacidal[33,34,37], we have demonstrated that these peptides enhance microglial activity generally and anti-fungal activity specifically. Future studies are required to understand the microglial receptors for Aβ peptides that mediate these responses and whether Aβ peptides induce similar anti-microbial responses from other cell types.

In addition to expressing Aβ peptides centrally and being comprised in part of highly activated DAMs that express CD68 and Dectin-1, FIGGs are further characterized by the presence of chitin centrally, demonstrating the presence of yeast cell aggregates. Remarkably, these are all features shared in common with senile plaques of AD[50,51]. Chitotriosidase, a mammalian enzyme that degrades chitin, which is not made by mammals, was also found to be upregulated in brains of AD patients[52] and may be a useful biomarker for AD[53].

Moreover, fungi and fungal components have also been detected in the peripheral blood[4,8] and cerebrospinal fluid[5,53] of AD patients. More extensive analysis specifically revealed the presence of *C. albicans* and other fungal species in the brains of AD patients, but not in healthy control brains[4,6,7]. Allergic asthma, which we have linked to airway mycosis, a form of superficial fungal infection of the airway mucosa[54], is epidemiologically linked to later onset dementia[55]. Our mice further developed memory deficits, another hallmark of AD, albeit a transient form that resolved with fungal clearance. AD is also associated with neuroinflammation marked by expression of NF-κB, IL-1β, IL-6, and TNF precisely as we observed[29].

Regardless of any possible link to AD, we have shown that transient fungemia in healthy mice has important physiological consequences, including alterations in working memory. Although transient after a single episode, it is conceivable that intermittent fungal showering of the vascular space and attendant low-grade fungal cerebritis that occurs over timeframes of years could eventually lead to permanent brain damage and lasting cognitive defects. More importantly, our findings suggest that resolution of low-grade CNS fungal infections through the use of antifungals and other means might preclude or even reverse attendant cognitive decline.

For long-term fungal bloodstream showering to occur, however, a peripheral site of chronic infection is also required. *C. albicans* and other *Candida* species comprise the normal microbial flora and such commensal organisms are unlikely to disseminate hematogenously[56]. However, multiple common medical practices conspire to alter the normal fungal microbiome or the immune system, leading to pathological enlargement of the human fungal biomass, often at mucosal sites. Such practices include the overuse of antibiotics, corticosteroids, hygiene products that disrupt potentially protective mucosal biofilms, and proton pump inhibitors that neutralize candidacidal stomach acid[16,57]. Consequently, the esophagus and more distal gastrointestinal tract may become highly colonized with *Candida* spp., a pathological condition that leads to systemic spread of the fungi[16,58,59] that in some cases leads to symptomatic fungal infections of the lung, liver, spleen, and kidneys[58,59]. Our studies

have shown that the CNS is also readily infected during low-grade candidemia, with important acute histologic and physiologic consequences.

The physiologic impact of chronic systemic fungal dissemination is not known, but the general importance of chronic organ inflammation due to unresolved infections and particulate, e.g., nanoparticulate carbon black derived from cigarette smoke[60], exposures is chronic inflammation and organ destruction that can be fatal if not checked. Thus, although a single low-grade challenge with *C. albicans* is quickly resolved and results in only transient physiologic derangement as shown here, the broader concern with chronic fungemia is diffuse end-organ injury, which in the CNS could include substantial neuronal loss and long-term, progressive cognitive impairment. Our findings thus support the creation of in vivo models that permit dissecting the impact of chronic candidemia on CNS integrity and function. Such models are likely to improve our understanding of chronic neurodegenerative conditions such as AD.

## Methods

**Mice.** Eight-week-old C57BL/6J male and female mice (wild-type, 5xFAD and homozygous $App^{-/-}$ mice) were purchased from Jackson Laboratories. All mice were bred and housed at the American Association for Accreditation of Laboratory Animal Care-accredited vivarium at Baylor College of Medicine under specific-pathogen-free conditions. All experimental protocols were approved by the Institutional Animal Care and Use Committee of Baylor College of Medicine, and followed federal guidelines.

**Sabouraud's agar plates and broth.** Sabouraud's agar or Sabouraud's broth (BD, Sparks, 21152) was dissolved in water at 50 g/l, and autoclaved (liquid, 30 min). Chloramphenicol (Sigma-Aldrich, St. Louis, 63103) was added to the solution at 50 mg/l. Sabouraud's broth was ready then, and the agar solution was sterilely poured onto Petri dishes (VWR, Radnor, 19087) and cooled overnight. Plates were sealed and kept at 4 °C for further experiments.

**Fungal isolation and maintenance.** Wild-type *C. albicans* was isolated from airway secretions of an asthma patient as described[61] and propagated on Sabouraud's agar plates. Mucoid colonies of *C. albicans* were harvested after growing to 10 mm diameter, and populated in Sabouraud's broth at 37 °C for 4 days. Cells were collected and dispersed in pyrogen-free phosphate buffered saline (PBS; Corning cellgro, Mediatech, Manassus, VA) passed through 40 μm nylon mesh, and washed twice with PBS by centrifugation ($10,000 \times g$, 5 min, 4 °C). Fungal cells were then suspended in PBS and aliquots frozen in liquid nitrogen at $5 \times 10^7$/ml. Viability after freezing (>95%) was confirmed by comparing hemacytometer-derived cell counts to CFU as determined by plating serial dilutions on Sabouraud's agar. Fungal identity was determined by standard morphology (Microcheck, Northfield, VT) as describe previously[14]. Thawed, >95% viable cells were washed once, counted, and suspended in normal saline at indicated concentrations for intravenous injection.

**Construction of fluorescent *Candida albicans*.** The pENO1-NEON-NAT[R] plasmid contains a codon-optimized version of the *NEON* gene under the control of the constitutive *ENO1* promoter and with a nourseothricin resistance marker (NAT[R]). Codon-optimized *NEON* (GenScrip Piscataway[62], NJ) was cloned into pENO1-dTomato[63] using NcoI and PacI, replacing the *dTomato* gene. The pENO1-NEON-NAT[R] plasmid was NotI-linearized within the ENO1 segment before transfection into *C. albicans* strain SC5314.

**Intravenous injection.** Viable cells of *C. albicans* in 100 μl normal saline were injected through the tail veil using a tuberculin syringe and 27-gauge needle. Mice are then returned to clean cages, and monitored carefully until resuming normal grooming.

**Brain dissemination assay.** Mice were euthanized with pentobarbital (Beuthanasia, Intervet Inc., Madison, NJ) and exsanguinated by transecting the descending aorta followed by perfusing the brain with normal saline. Brains were removed by sterilely removing the calvarium, weighed, and were put into 1 ml of sterile PBS. Brains were then homogenized, and spread directly onto Sabouraud's agar (one sample per plate). Plates were sealed with Parafilm (Pechiney Plastic Packaging, Chicago, IL) and incubated at 37 °C for a maximum of 10 days. CFU were enumerated and species confirmed as described above.

**Histology and immunostaining.** Mice were perfused with 4% PFA, and the brains were post-fixed in 4% PFA overnight at 4 °C followed by cryoprotection in 40% sucrose. Coronal sections of 30 μm were cut with a microtome and stored in a cryoprotectant at −20 °C. For each experiment, sections were collected randomly from at least three animals. Sections were mounted onto slides and were extensively washed in PBS, blocked with PBST containing 3% BSA and 2% donkey serum for 30 min, and then incubated in primary antibody diluted in blocking solution overnight at 4 °C (anti-GFAP (1:1000, Millipore, catalog #MAB3402), anti-Iba1 (1:500, Wako, catalog #019-19741), anti-NeuN (1:500, Millipore, Mab 377), anti-APP (1:250, Abcam, ab32136), anti-Aβ (1:250, Abcam, ab2539), anti-dectin-1 (1:100, Invivogen, #mabg-mdect), anti-CD68 (1:250, Biolegend, catalog #137001)). After washing, sections or coverslips were incubated in secondary antibody for 1 h at room temperature and mounted in DAPI solution after final washing.

**Calcofluor white stain.** Thirty micrometers of sections of brain on slides were treated with one drop of calcofluor white stain (18909, Sigma-Aldrich) and one drop of 10% potassium hydroxide for 1 min. The slide is then examined under ultraviolet light.

**Imaging.** Fluorescent immunostained brain sections were first imaged using the EVOS FL Auto system to locate sites of infection, and then were imaged using a Leica laser confocal microscope.

**Enumeration of brain microglia.** Immunostained coronal brain sections (30 μm) were fixed on slides and stained for IBA-1. Using a low-image threshold setting, total IBA-1-positive cells were first counted, after which IBA-1[high] cells were counted after raising the image acquisition threshold beyond which IBA-1[low] cells were no longer visible (ImageJ, ver. 1.51J8).

**Leukocyte isolation and single-cell suspension from brain.** Control and infected mice were euthanized 4 days post infection. Mice were anesthetized using pentobarbital (Beuthanasia, Intervet Inc.), perfused with normal saline and brains were collected in 3 ml of HBSS (Gendepot, Barker, TX) containing 20% FBS, and homogenized using the plunger portion of a 5 ml syringe in six-well flat bottom plates. The homogenate was then transferred to a 15 ml tube and added 1.25 ml of 90% Percoll (GE Healthcare) in PBS. Then, the suspension was underlaid with 3 ml of 70% Percoll and centrifuged at 2450 r.p.m. ($1200 \times g$) for 20 min at 4 °C. The leukocytes at the interphase were collected, washed with HBSS, and passed through a 40 μm filter[20].

**Cytokine measurement.** Following pentobarbital euthanasia, mice were perfused of blood as above. Brains were then harvested and deaggregated by pressing gently through 40 μm cell strainers in 3 ml of DMEM medium. The homogenate was then centrifuged ($1400 \times g$, 5 min, 4 °C), and supernatants were collected. IL-1β, IL-6, TNF, and amyloid β protein levels were detected using standard ELISA after comparison to recombinant standard (cytokines: ab210895, ab213749, and ab212073, Abcam, Cambridge, MA; amyloid β[64]: capture antibody: #800701, detection antibody: #805504 or #805404, 1:100, Biolegend, San Diego, CA).

**BV-2 fungus co-culture for ELISA.** BV-2 cells were originally obtained from ATCC and maintained frozen in liquid nitrogen. Cells were thawed, expanded in DMEM medium, and expression of standard microglial surface markers (TREM2, CD68, and MCM5 by real-time quantitative PCR) was confirmed. Cells were then seeded in 1 ml of DMEM (serum-free, Gendepot) in 24-well plates at 100,000 cells per well for 6 h. Live, irradiated *C. albicans* (200 cells/ml), *C. albicans* lysate (equivalent to 200 cells/ml), SAPs (1 μM), or protease activity-inhibited SAPs were then added to each well and incubated for 16 h at 37 °C. Supernatants were then collected for ELISA as described above.

**Preparation of *C. albicans* lysates.** *C. albicans* was cultured in suspension in liquid Sabouraud's broth (100 cc), enumerated by hemacytometer and collected by centrifugation. A planetary ball mill grinder (PM100, Retsch, Hann, Germany) was used to lyse the cells[54]. Briefly, pelleted fungal cells were resuspended in 30 ml by cold PBS, decanted into cold, sterile canisters and an equal volume of zirconium grinding beads, and centrifuged (550 r.p.m., 5 min, 3 cycles). The canisters were cooled on ice between each cycle for 5 min, the samples were removed, and centrifuged at 4000 r.p.m. ($3700 \times g$), 30 min, 4 °C in a separate centrifuge. The supernatants were transferred to a new tube and centrifuged again (8500 r.p.m./$6800 \times g$, 30 min, 4 °C), from which the supernatants were passed through 0.22 μm sterilizing filters, adjusted to a protein concentration of 6 mg/ml and distributed in 0.5 ml aliquots in sterile, 1 ml tubes for storage at −80 °C.

**Preparation of irradiated *C. albicans*.** *C. albicans* was cultured in suspension in liquid Sabouraud's broth (100 cc), enumerated by hemacytometer and collected by centrifugation. Cells were then irradiated using a [137]Cs irradiator (3000 rad, Gamma Cell 40, MSD Nordion, Ottawa, Ontario, Canada). Inactivation of *C. albicans* was confirmed by absent growth on Sabouraud's agar plate.

**Isolation of secreted aspartic proteinases**. SAPs were isolated as previously described[65,66]. Briefly, *C. albicans* was grown in YPD broth (BD, Sparks, 21152) for 24 h at 26 °C. Cells were removed by centrifugation (8500 r.p.m./6800 × *g*, 5 min, 4 °C) and the supernatants containing SAPs were concentrated 25 times in a Pierce Protein Concentrator (10 kDA MWCO, #88535, Thermo Fisher Scientific, Waltham, MA). Concentrated SAPs were then purified by passage through a Pierce Strong Anion Exchange Spin Column (#90011, Thermo Fisher Scientific, Waltham, MA). 20 mM Tris/HCl (pH 6.0) was used for column binding and 2 M Tris/HCl (pH 6.0) was used for elution. SAPs were then concentrated again as described above. SAP concentration was determined using a BCA protein assay kit (Thermo Fisher Scientific).

**Inhibition of secreted aspartic proteinase**. SAPs were incubated with halt protease and phosphatase inhibitor cocktail (#78442, Thermo Fisher Scientific) overnight. SAPs were then washed using 25 mM Tris/HCl (pH6.0) and concentrated as stated above three times to remove excessive inhibitors before applying to BV-2 cells. Absence of proteinase activity was confirmed by Coomassie Blue proteinase assay[65,66].

**Western blot and quantitative PCR**. Mice were euthanized and brains perfused and removed, after which they were either lysed using 1 ml of protein lysis buffer (50 mM NaCl, 20 mM HEPES, 1 mM EDTA, 2% Triton-X 100, 10% glycerol, with proteinase and phosphatase inhibitor) to extract total protein or 1 ml of TRIZol solution to extract RNA. NF-κB, Ikb-α, APP, beta secretase 1 (BACE-1), and presenilin 1 (PS1) levels were detected using western blot (4–12% Nupage Bis-Tris gel, Thermo Fisher Scientific) using standard technique (antibody: #8242S, #9242S, 1:1000, Cell signaling, Danvers, MA; ab 32136, 1:5000, Abcam; ab 2077, 1:1000, Abcam; MAB 5232, 1:1000, Millipore Sigma, Burlington, MA). Uncropped western blots in main figures are shown in Supplementary Figure 7. Relative expression of mRNA for APP was detected by two-step, real-time quantitative reverse transcription-PCR (RT-PCR) with the ABI Perkin Elmer Prism 5700 Sequence Detection System (Applied Biosystems, Foster City, CA) using Taqman probe (Mm00476361, Mm01344172, Invitrogen, Carlsbad, CA).

**In vitro fungistasis test using Aβ peptides**. Two hundred viable cells of *C. albicans* were cultured in 24-well flat tissue culture plates containing 1 ml of serum-free DMEM in a 37 °C/5% $CO_2$ incubator. Fungal germination events (FGEs) were counted after 16 h.

**In vitro fungistasis assay**. BV-2 cells were cultured in 24-well flat tissue culture plates at 12,000 cells per well for 6 h with stimulation by 1 μM Aβ peptides or scrambled peptide control (mouse Aβ: A-1007-1, A-1008-1, A-1004-1; human Aβ: A-1156-1, A-1166-1. rPeptide, Wadkinsville, GA[33]). *C. albicans* were added to each well at 200 viable cells per well in a 37 °C/5% $CO_2$ incubator. FGEs were counted after 16 h. Percent of fungal growth inhibition was calculated as the (# FGE in wells containing no cells − # FGE in wells containing cells/# FGE in wells containing no cells) × 100%. Cell-free supernatants from wells under these conditions were also transferred into new 24-well plates and 200 viable cells of *C. albicans* were added to each well and incubated for another 16 h. FGEs were counted and % FGE inhibition was calculated as above. In some experiments, BV-2 cells were first primed with 1 μM Aβ peptides or scrambled peptide control overnight in 24-well flat tissue culture plates at 12,000 cells per well. Cells were washed with fresh medium three times to remove Aβ peptides before adding *C. albicans*.

**Phagocytosis assay using BV-2 cells**. BV-2 cells were plated at $1 \times 10^6$/ml and stimulated with 1 μM Aβ peptides or scrambled peptide control for 6 h after which *C. albicans* (mNeonGreen) were added at $1 \times 10^6$/ml. BV-2 cells were co-cultured with *C. albicans* for 2.5 h in a 37 °C/5% $CO_2$ incubator, and washed with fresh medium three times. BV-2 cells were then collected using a cell scraper and stained with APC-Cy7-conjugated anti-mouse CD11b (M1/70, Biolegend). After three washes with PBS, cells were analyzed using ImageStreamX MKII (Millipore Sigma). CD11b/mNeonGreen double-positive cells were masked and imaged orthogonally to characterize fungal-BV-2 cell interactions as representing surface binding alone or true phagocytosis. Data were analyzed using FlowJo software (version 10.0.7; Treestar, Ashland, OR).

**Flow cytometry analyses**. Single-cell suspensions from brain were incubated at 4 °C for 30 min with APC-Cy7-conjugated anti-mouse Ly6C (RB6-8C5, BD Pharmingen, Franklin Lakes, NJ) and PE-conjugated anti-mouse CD11b (M1/70, eBioscience, San Diego, CA). BV-2 cells were incubated at 4 °C for 30 min with PE-Cy7-conjugated anti-mouse CD11b (M1/70, Biolegend). After three washes with PBS, flow cytometry was performed on an LSRII (BD Biosciences) and data were analyzed using FlowJo software (version 10.0.7; Treestar).

**Behavior tests**. Twenty-five thousand CFU or equivalent numbers of heat-inactivated cells of *C. albicans* were injected intravenously after which behavior tests were conducted 3 and 10 days later using different groups of mice. Open-field test, T-maze spontaneous alternation, and contextual fear conditioning was carried out in order to minimize the effect of stress as previously described[67].

**Open-field test**. Mice were placed in an open arena (40 × 40 × 20 cm) and allowed to explore freely for 10 min while their position was continually monitored using tracking software (AnyMaze). Tracking allowed for measurement of distance traveled, speed, and position in the arena throughout the task. Time spent in the center of the arena, defined as the interior 20 × 20 cm, was then recorded[67].

**T-maze spontaneous alternation task**. The apparatus was a black wooden T-maze with walls 25 cm high and each arm was 30 cm long and 9 cm wide. A removable central partition was used during the sample phase but not the test phase of each trial. Vertical sliding doors were positioned at the entrance to each goal arm. At the beginning of the sample phase, both doors were raised, and the mouse was placed at the end of the start arm facing away from the goal arms. Each mouse was allowed to make a free choice between the two goal arms; after its tail had cleared the door of the chosen arm, the door was closed, and the mouse was allowed to explore the arm for 30 s. The mouse was then returned to the end of the start arm, with the central partition removed and both guillotine doors raised, signaling the beginning of the test phase. Again, the mouse was allowed to make a free choice between the two goal arms. This sequence (trial) was repeated 10 times per day for 2 days. The percentage of alternation was averaged over the 2 days. Trials that were not completed within 90 s were terminated and disregarded during analysis[67].

**Contextual fear conditioning test**. Mice were first handled for 5 min for 3 days. On the training day, after 2 min in the conditioning chamber, mice received two pairings of a tone (2,800 Hz, 85 dB, 30 s) with a co-terminating foot shock (0.7 mA, 2 s), after which they remained in the chamber for an additional minute and were then returned to their cages. At 24 h after training, mice were tested for freezing (immobility except for respiration) in response to the training context (training chamber). Freezing behavior was hand-scored at 5-s intervals by an observer blind to the genotype. The percentage of time spent freezing was taken as an index of learning and memory[67].

**Statistical analyses**. Data are presented as means ± S.E. of the means. Significant differences relative to PBS-challenged mice or appropriate controls are expressed by *p* values of <0.05, as measured by two-tailed Student's *t*-test or one-way analysis of variance followed by Dunnett's test or Tukey's test for multiple comparison. Data normality was confirmed using the Shapiro–Wilk test.

## Data availability

The data that support the findings of this study are available from the corresponding author upon request.

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

## Acknowledgements

The authors thank Li-Zhen Song, Nima Baghaei, and Joel Sederstrom for technical assistance. C.T.L. is a Hudson Scholar in the Baylor College of Medicine Medical Scientist Training Program (Houston, TX, USA). Supported by US National Institutions of Health grants P30 AI036211, P30 CA125123, S10 RR024574; T32AI053831, R01HL117181, R01 AI135803, and R41AI124997; and VA Office of Research and Development grant 5I01BX002221.

## Author contributions

Y.W., F.K., H.Z., and D.B.C. designed the research, analyzed the data, and wrote the manuscript. Y.W., S.D., J.L.J., H.-Y.T., C.T.L., Y.L., B.G.S., R.T.W., and M.C.-M., performed the experiments and collected the data.

## Additional information

**Competing interests:** The authors declare no competing interests.

