## [Peer Review File · Nature Communications]

Reviewers' comments:

Reviewer #1 (Remarks to the Author):

In this paper, Wu et al describe a low-dose *Candida* infection that causes a transient brain invasion associated with amyloid precursor protein and microglial surrounding of *Candida*. The data are largely descriptive and do not provide direct evidence for the role of APP in host defense in the brain during this challenge. Data in *App*^{-/-} mice are surprisingly lacking. Below are some comments that may help the authors to enhance the conclusions:

1. Although the authors show with IF that APP and glial cells surround *Candida* in the infected brain, they present no data on how APP may affect (if it does) the host defense or behavioral parameters in mice. That is, there is no use of the *App*^{-/-} mice to gain insight on whether fungal burden is greater, whether IL-1, IL-6 or TNF are decreased, whether behavioral parameters are altered. Without that, it is difficult to know what APP does in vivo.
2. What happens to the number of microglia in the *Candida*-infected brain with the inoculum that the authors have used for infection? Do they increase in numbers? Are they activated when assessing activation markers?
3. In Figure 2H, it would be important to use *Sap*-deficient *Candida* to show that It does not induce cytokines from BV-2 cells to directly show that Saps (and not potential confounding *Candida* components) indeed drives cytokine production.
4. In Figure 3D, 3E and 4B it would be good to show CW stains as in the earlier Figure.
5. As above, in Figure 5A, it would be important to test *App*^{-/-} mice, not only the 5xFAD mice that may have baseline impaired brain immunity that could confound conclusions about the role of APP in host defense in vivo. Since the authors have access to the *App*^{-/-} mice they should thoroughly examine them in vivo in the brain after infection.
6. It is unclear what Fungistasis means. It is phagocytosis as shown in uptake experiments? It is killing? Stasis in microbiology would typically refer to inhibition of growth. Is this the measure of the assay?
7. The effects on phagocytosis are quite modest.

Reviewer #2 (Remarks to the Author):

In this manuscript, Wu and colleagues develop a new model of transient *Candida* Ceribritis and discover intriguing relationships between the focal accumulation of yeast within the brain, cognitive impairment, the glial response to *Candida*, and Amyloid Precursor Protein (APP)/ beta-amyloid. Their findings demonstrate a novel interaction between innate immunity and APP/beta-amyloid in the host response to pathogens in vivo and support the growing concept that APP-derived peptides could play important roles in host defense.

Overall this is a well-designed set of experiments and clearly written manuscript. The authors were also careful not to overstate or over interpret their findings with regard to the intriguing histological similarities between fungal-induced glial granulomas (FIGGs) and beta-amyloid plaques. However, for their findings to be more broadly applicable to human disease, some additional experiments and controls would be helpful to include:

1. It would very informative and considerably extend the importance of these findings if the authors can determine whether similar FIGG-like pathology can be detected in any human brain

samples. The authors point out that bloodstream candida infections are increasingly common and produce high mortality. Are any brain autopsy samples available from even just a few of these subjects? Alternatively or in addition, is there any evidence of candida-associated FIGGs in AD or MCI cases?

2. The BV2 phagocytosis assays demonstrate increased uptake of candida with co-exposure to beta-amyloid. However, it is not clear whether the amyloid is bound to the candida and helping the BV2 cells to recognize the pathogen or whether this is a more indirect effect mediated by the altered activation state of BV2 cells induced by beta-amyloid. Can red fluorescent Abeta be used in conjunction with the neon-green expressing candida to determine this?

3. Recent studies have shown that disease-associated microglia (DAMs) surrounding Abeta plaques in mouse models upregulate Clec7A/Dectin-1 (PMID: 28602351 and PMID: 28930663). Dectin-1 is a pattern recognition receptor that recognizes Candida beta-glucan. Therefore this recent finding has strong relevance to the current study. Do the authors observe similar upregulation of Dectin-1 in microglia associated with FIGGs? Can blocking or reduction of Dectin-1 in BV2 cells reduce Abeta-enhanced Candida phagocytosis?

4. A little data is shown from APP knockout mice that confirms the specificity of the APP histology around FIGGs. However, the authors mention that the "general appearance of FIGGs did not differ between wild type and app^{-/-} mice". Is the clearance of FIGGs delayed in APP Kos as would be expected given the putative role for APP and its peptide products in candida clearance? If so, then addition of Candida brain clearance data from this KO model in Fig5A would strengthen the authors' conclusions.

5. Does Abeta immunoreactivity persist once candida is cleared from the brain?

6. Although Fig 5A shows that 5xFAD mice clear candida infections more rapidly, is there any effect of candida on beta-amyloid load?

Reviewer #3 (Remarks to the Author):

SUMMARY

The present study investigated the impact of low-grade candidemia on its dissemination in the brain and pathological/behavioral outcomes. Intravenous injection of 25,000 C albicans significantly increased the number of yeasts in the brain within 4 days of challenge and formed FIGGs. Interestingly, NF- κ B, APP, BACE1, and PS1 were all upregulated in the brain between 4 and 14 days after the challenge, and immunofluorescent staining confirmed focal upregulation of APP around FIGGs and accumulation of Abeta within FIGGs. Wildtype mice with the low-grade challenge exhibited a transient impairment in T-maze test. In vitro BV-2 cell studies demonstrated that Abeta peptides induced potent anti-fungicidal activity of BV-2 cells via activating phagocytosis and secreting unknown anti-fungicidal molecules. Collectively, the present study nicely demonstrated the role of Abeta and microglia in low-grade candidemia in the brain and the induction of memory impairment in mice.

Overall, this study shows very interesting results. The critical role of Abeta in candida infection and its potential benefits and damage in the brain can be more convincing if complete data from APP-KO and 5xFAD mice are available. Findings from these mice will give us whether deficiency or overexpression of APP modifies the severity of infection in the brain, and whether subsequent transient memory impairment is due to increased accumulation (or production) of Abeta in the brain as a result of infection. More detailed evaluation of the potency and property between mouse

and human Abeta species would also strengthen the current findings. Below is the list of specific concerns.

MAJOR CONCERNS

1. Did mouse or human Abeta peptides (e.g. Abeta40, Abeta42) significantly increase following the challenge in wildtype and 5xFAD mice? Are these accumulated Abeta on FIGG soluble or insoluble forms of Abeta?
2. Does APP-KO mouse have greater number of yeast and FIGG in the brain and/or take longer time to clear them from the brain after the challenge compared to wildtype or 5xFAD due to lack of APP? This information could further support the role of Abeta in triggering potent fungicidal activity in immune cells.
3. Also, does APP-KO mouse exhibit more cognitive decline following the challenge compared to wildtype mice?
4. Please indicate if human or mouse Abeta peptides were used in fungicide and fungistasis tests. If human Abeta peptides were used, do mouse Abeta exhibit similar results as well? Please also discuss if the concentration (2 ug/ml) of Abeta peptides was physiologically-relevant.
5. To better link in vitro and in vivo findings, please show evidence of microglial phagocytosis of yeast around FIGG. Did phagocytosis markers (such as YM1, Arg1, etc) or phagocytosed yeasts (e.g. CD68/yeast colocalization within microglia) were significantly increased in microglia surrounding FIGG?
6. Increased expression or protein levels of NF-kB does not necessarily mean the increased activation of NF-kB. Any changes in IκB levels or NF-kB nuclear translocation?

MINOR CONCERNS

1. Please clarify if BV-2 cells were purchased from ATCC as indicated in the method. Please include catalog #.

Monday, October 1, 2018

Re: NCOMMS-18-08484 (Microglia and Amyloid Precursor Protein Coordinate Control of Transient Candida Cerebritis With Memory Deficits) Response to Reviewers.

1. Reviewer #1

1. *Although the authors show with IF that APP and glial cells surround Candida in the infected brain, they present no data on how APP may affect (if it does) the host defense or behavioral parameters in mice. That is, there is no use of the App^{-/-} mice to gain insight on whether fungal burden is greater, whether IL-1, IL-6 or TNF are decreased, whether behavioral parameters are altered. Without that, it is difficult to know what APP does in vivo.*

Response: We agree that studying more intensively app^{-/-} mice in the *C. albicans* model is an obvious course, but was one we could not pursue as part of our original submission. In the interim, we were able to secure additional mice and have conducted the suggested experiments, now demonstrating that app^{-/-} mice are unable to control cerebral *C. albicans* as well as wild type and to a much less degree as compared to 5xFAD mice at least through day 7; these mice further mount reduced IL-1 β and TNF responses over 4-7 days (Fig 5A, Suppl Fig 5). Of note, app^{-/-} mice were overall much sicker than wildtype mice—they were hypothermic, experiencing frequent rigors and lethargy, behavioral changes that precluded our ability to rigorously assess transient cognitive changes.

2. *What happens to the number of microglia in the Candida-infected brain with the inoculum that the authors have used for infection? Do they increase in numbers? Are they activated when assessing activation markers?*

Response: We agree that quantification of microglia is of importance. IBA-1 is considered a sensitive marker for microglia and differs greatly in surface expression between surveilling and activated microglia. Thus, we have now differentiated and quantified microglia in terms of their different activation states in the coronal sections stained for IBA-1. These new data clearly demonstrate the enormous upregulation of activated microglia during *C. albicans* cerebritis (Suppl Fig 2).

-Ito D, Imai Y, Ohsawa K, et al. Microglia-specific localisation of a novel calcium binding protein, Iba1[J]. *Molecular brain research*, 1998, 57(1): 1-9.

-Lynch M A. The multifaceted profile of activated microglia[J]. *Molecular neurobiology*, 2009, 40(2): 139-156.

3. *In Figure 2H, it would be important to use Sap-deficient Candida to show that It does not induce cytokines from BV-2 cells to directly show that Saps (and not potential confounding candida components) indeed drives cytokine production.*

Response: We agree with this suggestion and therefore purified SAPs using anion exchange column chromatography and also carried out a protease inhibition control in which we inhibited all SAPs prior to stimulation of BV-2 cells. This treatment abolished the activation effect, clearly demonstrating the importance of SAPs for cytokine secretion from BC2 cells (Fig 2H).

4. *In Figure 3D, 3E and 4B it would be good to show CW stains as in the earlier*

Figure.

Response: We have included more CW staining, verifying that FIGGs are tightly associated with fungal clusters. Please note that given the caustic nature of the calcofluor white reagent that rapidly degrades brain slices, this stain could not be performed for all experiments.

5. *As above, in Figure 5A, it would be important to test App^{-/-} mice, not only the 5xFAD mice that may have baseline impaired brain immunity that could confound conclusions about the role of APP in host defense in vivo. Since the authors have access to the App^{-/-} mice they should thoroughly examine them in vivo in the brain after infection.*

Response: We have included these experiments and were discussed above.

6. *It is unclear what Fungistasis means. It is phagocytosis as shown in uptake experiments? It is killing? Stasis in microbiology would typically refer to inhibition of growth. Is this the measure of the assay?*

Response: The fungistasis assay measures the inhibition of growth of fungus *in vitro* although we have also developed an *in vivo* fungistasis assay for lung. Typically, it measures the inhibition of fungal growth mediated by cells after different stimuli, which is the combination of both secretory inhibition and phagocytosis (Fig 5D). Taking the supernatant and the cellular component apart will further indicate the role of secretory inhibition (Fig 5E) and phagocytosis (Fig 6). We have included a new schematic diagram for this assay (Fig 5B) and a more detailed description of the fungistasis assay in the manuscript.

7. *The effects on phagocytosis are quite modest.*

Response: We have utilized the ImageStreamX Mark II flow cytometer to more rigorously quantify phagocytic events, yielding more robust data (Fig 6).

2. Reviewer #2

1. *It would very informative and considerably extend the importance of these findings if the authors can determine whether similar FIGG-like pathology can be detected in any human brain samples. The authors point out that bloodstream candida injections are increasingly common and produce high mortality. Are any brain autopsy samples available from even just a few of these subjects? Alternatively or in addition, is there any evidenced of candida-associated FIGGs in AD or MCI cases?*

Response: We have to be careful in this initial description of our work in a new field to not draw too strong a comparison to chronic disorders such as AD where a potential link to infectious etiologies is only now emerging, and is highly controversial. Nonetheless, we more clearly point out in the discussion that every feature of FIGGs, including the presence of chitin, has already been shown to occur in senile plaques of AD and cite the relevant literature. We will address as part of a future manuscript the potential direct link between fungi and senile plaques.

2. *The BV2 phagocytosis assays demonstrate increased uptake of candida with co-exposure to beta-amyloid. However, it is not clear whether the amyloid is bound to the candida and helping the BV2 cells to recognize the pathogen or whether this is a more indirect effect mediated by the altered activation state of BV2 cells induced by beta-amyloid. Can red fluorescent Abeta be used in conjunction with the neon-green expressing candida to determine this?*

Response: We agree that this is an important distinction to make and have modified our fungistasis assay by first priming the BV-2 cells with A β before adding A β -free cells to the fungistasis assay (Fig 5B). We now show that BV-2 cells show enhanced fungistatic activity under these conditions (Fig 5F). We are currently working to demonstrate the specific A β receptor on BV-2 cells that mediates this effect, but these studies are beyond the scope of this manuscript.

3. *Recent studies have shown that disease-associated microglia (DAMs) surrounding Abeta plaques in mouse models upregulate Clec7A/Dectin-1 (PMID: 28602351 and PMID: 28930663). Dectin-1 is a pattern recognition receptor that recognizes Candida beta-glucan. Therefore this recent finding has strong relevance to the current study. Do the authors observe similar upregulation of Dectin-1 in microglia associated with FIGGs? Can blocking or reduction of Dectin-1 in BV2 cells reduce Abeta-enhanced Candida phagocytosis?*

Response: This was a fascinating suggestion from the reviewer that we looked closely into. We now include data showing that the microglia around FIGGs are highly activated and indeed express Clec7A/Dectin-1 (Fig 7A), consistent with them being DAMs. We also determined the reduced phagocytosis of BV-2 cells post Dectin-1 blockage (Fig 7B-D).

4. *A little data is shown from APP knockout mice that confirms the specificity of the APP histology around FIGGs. However, the authors mention that the “general appearance of FIGGs did not differ between wild type and app-/- mice”. Is the clearance of FIGGs delayed in APP Kos as would be expected given the putative role for APP and its peptide products in candida clearance? If so, then addition of Candida brain clearance data from this KO model in Fig5A would strengthen the authors conclusions.*

Response: We agree with the reviewer and have completed the suggested experiments (Fig 5A, Suppl Fig 5). These data show clearly that at early time points, rapidity of clearance of *C. albicans* from mouse brains is directly dependent on the amount of brain APP present.

5. *Does Abeta immunoreactivity persist once candida is cleared from the brain?*

Response: Yes. We have indicated this point in the revised manuscript and this is further supported by the upregulation of APP and A β even after fungal clearance at day 10 (Fig 3B and 4C, D).

6. *Although Fig 5A shows that 5xfAD mice clear candida infections more rapidly, is*

there any effect of candida on beta-amyloid load?

Response: Yes. We now show (Fig. 4C,D) that *C. albicans* cerebritis induces the highly significant upregulation of both A β 1-40 and 1-42 peptides in the brain and that these peptides remain elevated even 4 days after resolution of the infection.

3. Reviewer #3

1. *Did mouse or human Abeta peptides (e.g. Abeta40, Abeta42) significantly increase following the challenge in wildtype and 5xFAD mice? Are these accumulated Abeta on FIGG soluble or insoluble forms of Abeta?*

Response: As above, we confirmed the upregulation of both peptides by ELISA (Fig 4C, D).

2. *Does APP-KO mouse have greater number of yeast and FIGG in the brain and/or take longer time to clear them from the brain after the challenge compared to wildtype or 5xFAD due to lack of APP? This information could further support the role of Abeta in triggering potent fungicidal activity in immune cells.*

Response: As above, we have performed the suggested experiments (Fig 5A, Suppl Fig 5).

3. *Also, does APP-KO mouse exhibit more cognitive decline following the challenge compared to wildtype mice?*

Response: Unfortunately, and in contrast to wild type, *C. albicans* infected *app*^{-/-} mice manifest obvious morbidity. These animals are visibly tremulous and much less active, precluding obtaining accurate data from behavioral tests. We are exploring remedies for this situation and hope to report on this in a subsequent publication.

4. *Please indicate if human or mouse Abeta peptides were used in fungicide and fungistasis tests. If human Abeta peptides were used, do mouse Abeta exhibit similar results as well? Please also discuss if the concentration (2 ug/ml) of Abeta peptides was physiologically-relevant.*

Response: We now specify more clearly where we used mouse A β peptides for our fungistasis assay and where we used human A β (Suppl Fig 6). We corrected the concentration of A β to 1 μ M (~4 μ g/ml) and discussed the physiological relevance in the discussion.

5. *To better link in vitro and in vivo findings, please show evidence of microglial phagocytosis of yeast around FIGG. Did phagocytosis markers (such as YMI, Arg1, etc) or phagocytosed yeasts (e.g. CD68/yeast colocalization within microglia) were significantly increased in microglia surrounding FIGG?*

Response: We have included staining of CD68 on microglia around FIGGs (Fig 6E). Moreover, we also stained for Clec7a/Dectin-1 which marks Disease-Associated Microglia and can indicate phagocytosis of fungus (Fig 7A).

- Herre J, Marshall A S J, Caron E, et al. Dectin-1 uses novel mechanisms for yeast phagocytosis in macrophages[J]. *Blood*, 2004, 104(13): 4038-4045.

- Keren-Shaul H, Spinrad A, Weiner A, et al. A unique microglia type associated

with restricting development of Alzheimer's disease[J]. Cell, 2017, 169(7): 1276-1290. e17.

6. *Increased expression or protein levels of NF- κ B does not necessarily mean the increased activation of NF- κ B. Any changes in I κ B levels or NF- κ B nuclear translocation?*

Response: We have now determined the protein levels of I κ B- α from brains post infection (Suppl Fig 3).

Minor concern 1. Please clarify if BV-2 cells were purchased from ATCC as indicated in the method. Please include catalog #.

Response: We now specify that we obtained the BV-2 cells from the ATCC, although this cell line has since been discontinued. Microglial properties of this cell line have been confirmed by qPCR with respect to expression of TREM2, CD68, and MCM5.

David B. Corry, M.D.
(for the authors)

REVIEWERS' COMMENTS:

Reviewer #1 (Remarks to the Author):

The authors have satisfactorily addressed my comments in this revised manuscript.

Reviewer #2 (Remarks to the Author):

The author's have carefully address my prior critiques and the manuscript is substantially improved. This study provides a fascinating new set of findings that greatly improve our understanding of CNS candida cerebritis. In addition, these findings provide intriguing new data that further supports the growing realization that infectious diseases and the brain's response to infections likely plays an important role in Alzheimer's Disease.

Reviewer #3 (Remarks to the Author):

The authors sufficiently addressed most of concerns raised by the reviewers. New data are now included, and their findings and conclusion appear to be solid. No further concern is noted.